# Are the forearm muscles excited equally in different, professional piano players?

**Alba Thio-Pera**[1]◎*, **Matteo De Carlo**[2,3], **Andrea Manzoni**[4], **Fabrizio D'Elia**[4], **Giacinto Luigi Cerone**[1,2], **Giovanni Putame**[2,3], **Mara Terzini**[2,3], **Marco Gazzoni**[1,2], **Cristina Bignardi**[2,3], **Taian Vieira**[1,2]◎

**1** Laboratory for Engineering of the Neuromuscular System (LISiN)—Politecnico di Torino, Turin, Italy, **2** Polito^BIO^Med Lab, Politecnico di Torino, Turin, Italy, **3** Department of Mechanical and Aerospace Engineering, Politecnico di Torino, Turin, Italy, **4** Another Music Records, Paris, France

◎ These authors contributed equally to this work.
* thioalbs@gmail.com

**Data Availability Statement:** All relevant data files are available from the Figshare database (DOI: 10. 6084/m9.figshare.18753014).

## Abstract

### Background and objectives

Professional pianists tend to develop playing-related musculoskeletal disorders mostly in the forearm. These injuries are often due to overuse, suggesting the existence of a common forearm region where muscles are often excited during piano playing across subjects. Here we use a grid of electrodes to test this hypothesis, assessing where EMGs with greatest amplitude are more likely to be detected when expert pianists perform different excerpts.

### Methods

Tasks were separated into two groups: classical excerpts and octaves, performed by eight, healthy, professional pianists. Monopolar electromyograms (EMGs) were sampled with a grid of 96 electrodes, covering the forearm region where hand and wrist muscles reside. Regions providing consistently high EMG amplitude across subjects were assessed with a non-parametric permutation test, designed for the statistical analysis of neuroimaging experiments. Spatial consistency across trials was assessed with the Binomial test.

### Results

Spatial consistency of muscle excitation was found across subjects but not across tasks, confining at most 20% of the electrodes in the grid. These local groups of electrodes providing high EMG amplitude were found at the ventral forearm region during classical excerpts and at the dorsal region during octaves, when performed both at preferred and at high, playing speeds.

### Discussion

Our results revealed that professional pianists consistently load a specific forearm region, depending on whether performing octaves or classical excerpts. This spatial consistency may help furthering our understanding on the incidence of playing-related muscular

**Funding:** The author(s) received no specific funding for this work.

**Competing interests:** The authors have declared that no competing interests exist.

disorders and provide an anatomical reference for the study of active muscle loading in piano players using surface EMG.

## Introduction

Piano playing techniques demand a fast coordination of intellectual and body responses to accomplish the specific speed, timber and tempo required by each musical piece. To ensure the successful execution of a musical piece, pianists, as well as other musicians, often engage in several hours of practice, which may lead to overuse of the body parts involved. Moreover, it has been shown that from 60% to 87% of professional musicians will, at some point of their career, develop a playing-related musculoskeletal disorder (PRMD) [1–4]. More specifically, for professional piano players, the prevalence of musculoskeletal related pain or injuries goes from 50% to 70% [5, 6]. The incidence of these disorders for piano players is reported mostly for the wrist (30.6%-53.2%), forearm (24%), shoulder (20%-35%) and hand (22.6%) [7, 8]. Although injuries may also manifest in the neck and back, these are seemingly more related to prolonged maintenance of harmful body posture than to piano playing itself. For piano playing, different studies have documented an association between the occurrence of musculoskeletal disorders and playing aspects, such as hand span, number of practice hours per week and joint angles of wrist and elbow while playing the piano [9]. Furthermore, different techniques have been reported for different players, some of them leading to a more efficient muscle activation and energy consumption [10]. Thus, playing strategies and active muscle loading appear to impact on the probability of developing musculoskeletal injuries.

Different attempts proposed to study the association between muscle loading and the possible causes of PRMD in pianists rely on the assessment of excitation of forearm muscles from surface electromyograms (EMGs). For instance, higher muscle excitation has been observed when playing louder or faster [11–13]. Moreover, experience in piano playing has been reported to markedly affect both the pattern of muscle excitation and the biomechanics of movement, with skilled players exhibiting less co-activation of extrinsic finger extensors before key-strike and greater elbow pronation-supination and finger attack angles when compared with unskilled pianists sit [14]. A common feature shared by these studies is that surface EMGs have been collected locally, with a single pair of electrodes from specific, forearm regions. The local sampling of EMGs is not necessarily an issue per se, but it implicitly presumes the signals collected are sensitive to variations in muscle excitation taking place during the piano performance. It further assumes the EMG sensitivity to changes in muscle excitation is subject invariant. However, at least during isometric contraction of individual finger extensors, the sampling of EMGs with a grid of electrodes suggests excitation may take place in different forearm regions [15, 16]. Benefiting from this high-density surface EMG detection [17], Goubault et al have recently reported the manifestation of early signs of muscle fatigue to be localized mainly in wrist and finger extensors in expert piano players [18]. It seems therefore plausible to expect that specific forearm regions may be distinctively sensitive to variations in muscle excitation during piano playing across subjects. If this is the case, it would seem possible to expose these most sensitive regions with HD-EMG.

In this study we therefore use high-density EMG to investigate the regional excitation of forearm muscles during piano playing. We specifically ask whether, with respect to anatomical landmarks, there are specific forearm regions where EMGs with greatest amplitude are more likely to be detected when expert piano players perform different excerpts. We address this

issue for the forearm muscles because musculoskeletal injuries are frequently reported for the wrist and forearm in piano players [8, 9] and because there are several abutting muscles in the forearm, making it difficult to isolate individual muscles from the skin surface. Considering overuse is the likely cause of the typically reported injuries in piano players, we expect to identify common regions across subjects where greatest EMGs are to be detected. By quantifying these regions in relation to anatomical landmarks, we further expect to provide a reference location for the study of muscle function from surface EMG in piano players.

## Methods

Eight (2 females) professional piano players volunteered to participate in this study (range; age: 23–41 years; height: 164–183 cm; body mass: 56–88 kg). Participants were instructed about the experimental procedures to be undertaken and experiments commenced after written, informed consent was obtained. All subjects were expert players, defined according to the minimum time of professional playing (>15 years) and all practiced, on average, over 4 hours a day. None reported to have had any history of musculo-skeletal impairments or pain impairing their musical performance. The experimental protocol conformed with the Declaration of Helsinki and was approved by the Regional Ethics Committee (Commissione di Vigilanza, Servizio Sanitario Nazionale—Regione Piemonte— ASL 1—Torino, Italy).

## Experimental protocol

Before the session started, musicians were given a few minutes to warm-up. Then, they were asked to perform 18 tasks with any necessary breaks in-between. Subjects were asked to sit comfortably on a chair of adjustable height, ensuring participants could perform at their preferred posture. The music score of each task along with specific instructions on the task to be performed were sent to the participants at least five days prior to the experiments, ensuring all would be equally acquainted with the tasks (S1 Appendix).

Tasks were divided into two groups. For the first group, subjects were requested to perform the seven octaves sequentially, from C1-B1 to C7-B7, using the thumb and little fingers of the right hand alternately. Octaves were performed three times for each of the following four conditions: spezzate, forearm, wrist and fingers. In the latter three conditions, subjects were asked to respectively load mainly the forearm, the wrist and the fingers segments, each at a time. For the spezzate, forearm and fingers conditions, the wrist was kept at the neutral position [19]. Considering the effect of movement speed on the excitation of forearm muscles during piano playing [20], subjects were asked to perform the octaves at a self-paced speed and as fast as possible. A total of 24 octave trials were conducted per subject (4 conditions x 2 speeds x 3 repetitions).

The second group of tasks comprised six, classical music excerpts played at the preferred tempo: 1) Prelude in C-sharp minor, Opus 3 n.2 of Rachmaninov, 2) Impromptu Op. 90 No. 2 of Schubert, 3) Schicksal in Arbait of Andrea Mazoni [1] and 4) improvisation during at least three minutes. Each classical performance was repeated three times, providing a total of 10 trials (3 repetitions x 3 classical music excerpts and the improvisation trial). Octaves and these pieces of classical music were selected for two reasons. First, random playing has been shown to demand different degrees of muscle excitation when compared to sequential playing [20]. Second, this selection would allow us to account for the presumably large variation of movements imposed by the playing exercises and by classical music performance when testing the hypothesis of a consistent distribution of muscle excitation across subjects and conditions.

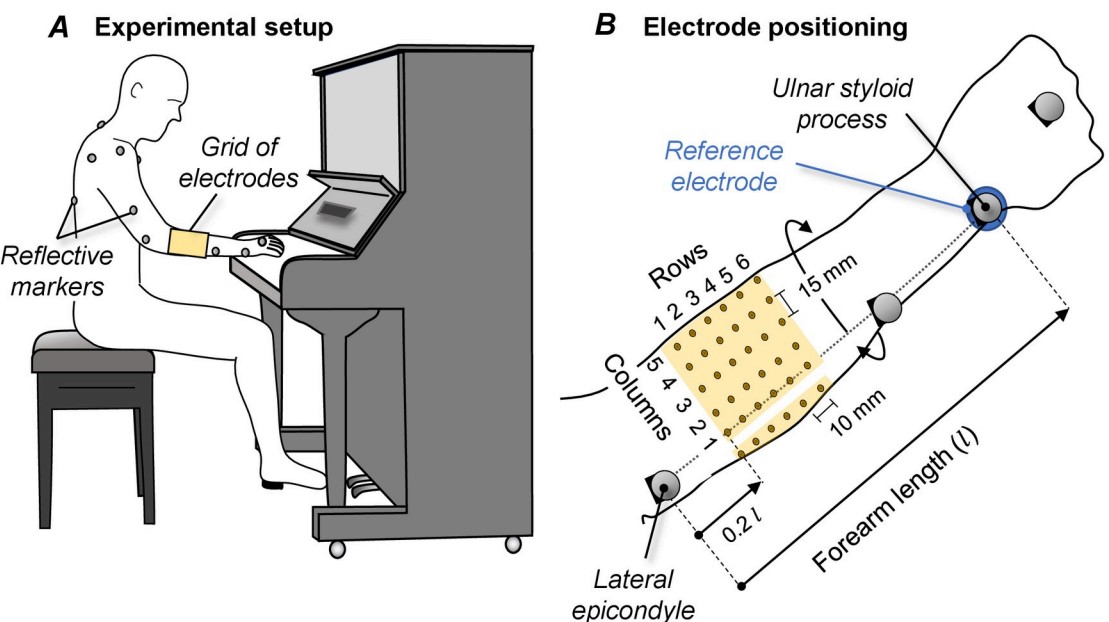

**Fig 1. Experimental setup.** A. Schematic representation of the experimental protocol and of the positioning of markers and of the grid of electrodes. The detailed procedure considered for standardizing the position of the grid of electrodes across participants is illustrated in B. The first column of electrodes was placed along the line connecting the lateral epicondyle to the ulnar styloid process, whereas 20% of this line was the reference for placing the first row of electrodes.

## Instruments

All subjects performed on a vertical piano (Fig 1A; U1, Yamaha, Japan). Given differences in key stiffness have been shown to impose different muscle demands [21], we decided to use an instrument for which keystrokes would be felt as if subjects were engaged in a classical music performance.

## Surface electromyography

Monopolar EMGs were collected with three high-density grids of 32 electrodes each. Electrodes were arranged into 2 rows and 16 columns, with 10 mm and 15 mm inter-electrode distances (IEDs) along rows and columns respectively. First, anatomical measurements were made to assist in the positioning of electrodes. The line connecting the lateral epicondyle to the ulnar styloid process was drawn and its length was measured, with the forearm resting on a table on pronated position and the elbow joint flexed at 90º. The length of this line was deemed to be the forearm length (Fig 1B). The circumference of the forearm was measured at 20% of its length, starting from the lateral epicondyle. We used these anatomical references for standardizing the position of the grids of electrodes across subjects, according to results reported in a previous study during isometric, voluntary contractions [15]. Specifically, columns of electrodes were aligned transversely to the forearm longitudinal axis, with the bottom row of electrodes of the most proximal grid being placed at 20% of the forearm length, over the forearm circumference (Fig 1B).

Before positioning electrodes, the skin was shaved and cleaned with abrasive paste. A 1 mm thick, double adhesive pad was used to secure the grids to the skin after filling, with conductive paste, the holes punched at the pad in correspondence of the electrodes. A single pad was used

to secure the three grids to the skin, ensuring the same IED (10 mm) along the forearm both within and between grids. The reference electrode was placed at the medial epicondyle.

Surface EMGs were recorded with a miniaturized, wireless, modular system [22] (LISiN-Politecnico di Torino, Italy). Signals were amplified by a factor of 200 and then digitized at 2048 samples/s using a 16bit A/D converter.

## Motion capture

Three-dimensional motion data was captured by 12 infrared cameras (Vero v2.2, Vicon system, Oxford, UK). Nineteen reflective markers were placed with double-sided adhesive tape at specific, anatomical landmarks, following a modification of the full-body Plug-in Gait Reference model (Fig 1 from Nexus Software v2.11, Vicon, Oxford, UK). Seven different segments were modeled: torso and arm, forearm and hand, bilaterally. The coordinates of each marker in the laboratory reference frame were sampled at 100 Hz.

A TTL signal, issued by the motion capture system, was sent to each of the three modules for the acquisition of EMGs via a radio-frequency transmitter. This signal was then sampled concurrently with EMGs, allowing for the offline synchronization of HD-EMGs and kinematics data.

## Data analysis

The first step of the data analysis consisted in assessing the quality of EMGs through visual inspection. Monopolar EMGs presenting power line interference (>20 uVpp), movement artefacts superimposed on periods of muscle excitation or contact problems [17] were replaced using a linear interpolation of the neighbor channels. Then, EMGs were band-pass (20–400 Hz) filtered with a 4th order, zero-lag Butterworth filter.

When collecting high-density EMGs to assess muscle excitation, interest typically lies on three aspects: where, how much and to what spatial extent the muscle is excited. This information may be retrieved from the amplitude of surface EMGs detected from multiple, skin locations (cf. Fig 4 in Vieira and Botter [17]). Accordingly, here we are interested in verifying whether, across subjects, there are specific forearm regions where excitation is most likely to occur. We therefore started by computing the Root Mean Square (RMS) amplitude of each monopolar EMG in the grid. For the octaves, RMS values were computed over periods corresponding to both left and right movements of the right wrist marker and then averaged across up-down and left-right cycles, separately for the self-paced and fast playing speeds (Fig 2). The 5th and 90th percentile between consecutive, local maxima and minima in the right wrist vertical position were considered to identify down phases and vice-versa for identifying up, cycle phases. With this procedure we aimed to remove transitory effects, associated with any short rest periods or with changes in movement direction. For the classical music playing, RMS values were computed over the whole performance and averaged across the three repetitions of the same piece. We acknowledge the classical performance presumably imposes high, non-stationarities in the signals. Nevertheless, we expect the forearm regions providing frequently high EMG amplitude to be represented in the RMS image computed over the whole, classical playing.

In order to account for the effect of anatomical differences between subjects, we normalized the position of electrodes with respect to the forearm circumference and length. More specifically, rows and columns were respectively scaled to a percentage of the forearm length and circumference. The smallest covered region in the longitudinal and transverse, forearm directions, was taken as a reference for defining the largest skin region to be analyzed across all subjects. That is, grids were truncated and linearly interpolated for subjects with covered

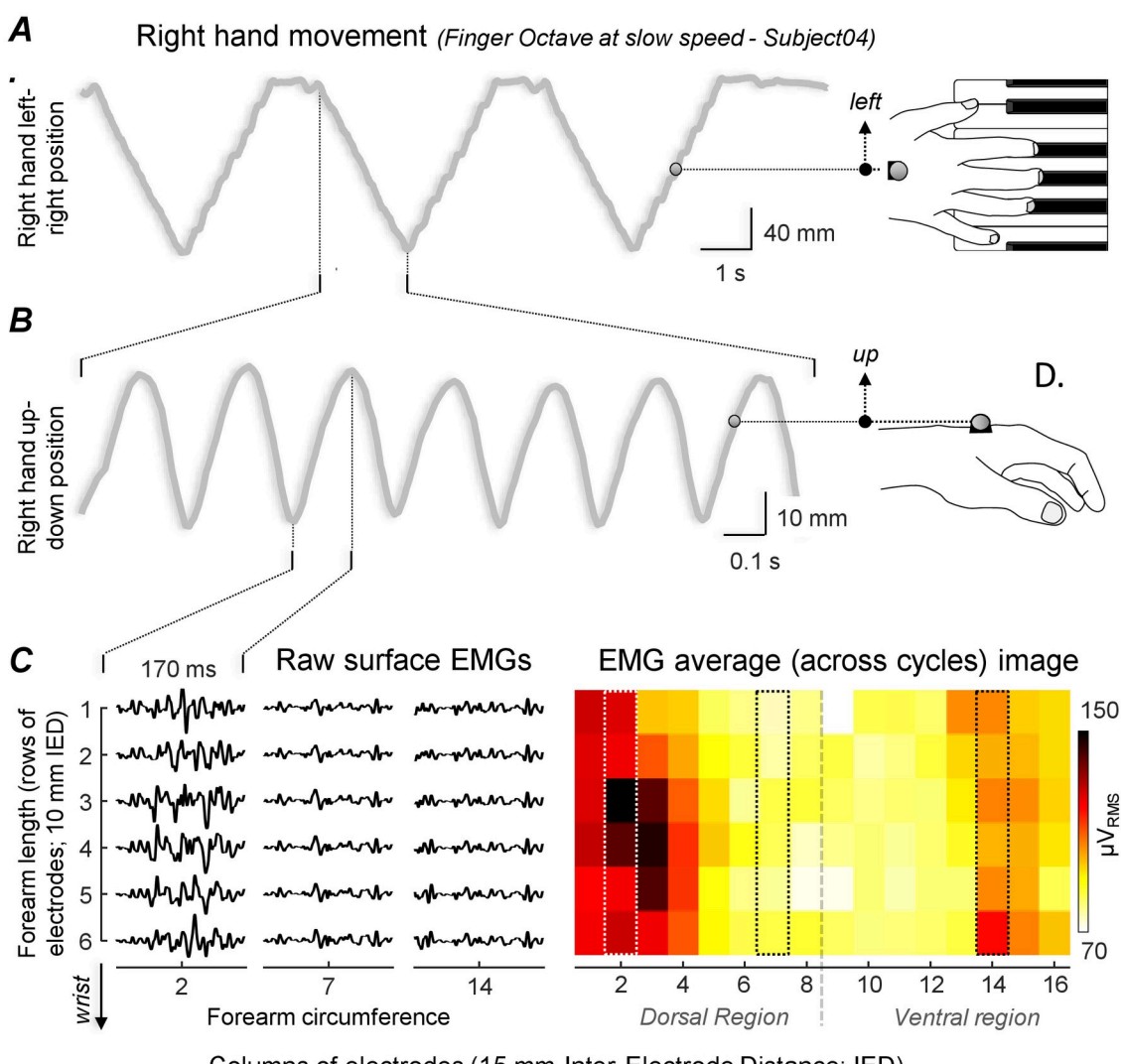

**Fig 2. Computation of EMG maps for octaves.** Hand trajectories along the lateral and the craniocaudal axes are illustrated respectively in panels A and B for a representative subject during wrist octaves. Raw EMGs detected by all electrodes in columns 2, 7 and 14 during an upward phase of the up-down hand trajectory cycles are shown in panel C (left), together with the Root Mean Square (RMS) amplitude of each monopolar EMG averaged across cycles (right).

regions larger than the reference region, providing an equal number of rows and columns for all subjects and an anatomical reference for the mapping of EMG amplitudes. For both octaves and classical music performances, we computed RMS images over 1 s of rest. Difference, RMS maps were computed, separately for rest and piano playing conditions (cf. Step 1 in Fig 3). Images were normalized with respect to their average RMS value and subtracted: normalized RMS image during piano playing minus normalized RMS image during rest. From these difference images we assessed the presence of zones providing EMGs with consistently high amplitude across subjects.

## Statistics

A non-parametric permutation test was used to identify common regions of excitation across subjects, following an adaptation of the third practical example explained by Nichols et al. [23].

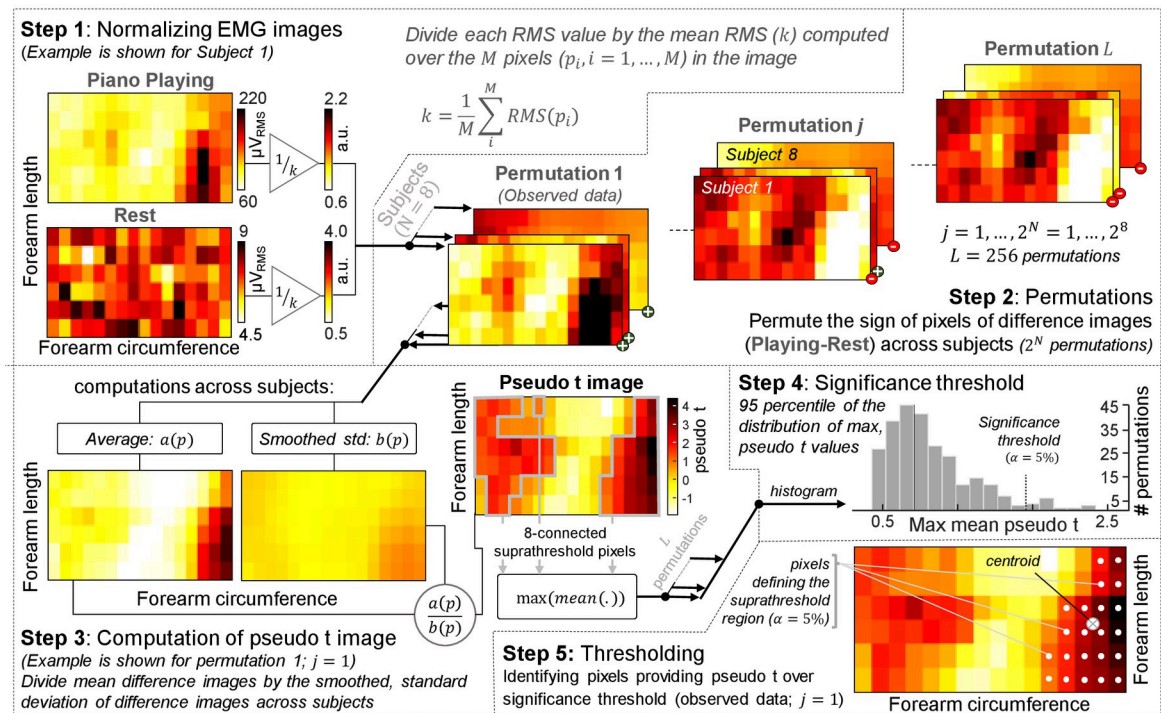

**Fig 3. Statistical assessment of consistent excitation across subjects.** This figure systematically illustrates the procedures necessary for applying the statistical, permutation test described by Nichols et al. [23] to EMG images. The method is spited into five steps, detailed in the text and summarized in insets. In all images illustrated, higher and lower values are represented with respectively darker and brighter pixels. The example is shown for the improvisation task.

Fig 3 systematically illustrates the procedures requested for applying this inferential, statistical test to EMG images. Briefly, under the null hypothesis that there is no common region of excitation, the value of every single pixel in the difference maps would be distributed symmetrically about zero across subjects. Considering the reasonable assumption of independence between subjects, flipping the sign of the observed data of any or all subjects would not affect the joint distribution of the whole data: pixel values would still be distributed symmetrically about zero (exchangeability of subjects [23]). Then, we considered two possible labels, + and -, respectively determining whether to flip or not to flip the sign of all pixel values in the image and thus providing a total of $L = 256 = 2^{8 \text{ subjects}}$ possible permutations (Fig 3, Step 2). For each permutation and for each pixel in the difference images, we computed the mean and the standard deviation value across subjects, respectively producing average and standard deviation images. The standard deviation image was smoothed by replacing the value of each pixel with the average of its surrounding values, accounting for spurious occurrences of near-zero variance. By dividing the average image by the smoothed, standard deviation image we obtained images with pseudo t values—the use of a smoothed, standard deviation is what differentiates the pseudo t value from the standard t value. From these images we computed the test statistics according to the suprathreshold cluster approach [23]. First, we set the median of the pseudo t map as the primary threshold and identified all suprathreshold clusters with 8-connected pixels (Fig 3, Step 3). Then, we calculated the mean value of each suprathreshold clusters and retained the maximal, mean value. After that, we defined the significance threshold as the 95 percentile of the distribution of maximal, suprathreshold clusters mean values across the 256 permutations (Fig 3, Step 4), controlling for image-wise Type I error [24]. Finally, we used this threshold to

identify suprathreshold clusters associated with significant, mean values in the observed data (pseudo t image for permutation 1; Fig 3, Step 5). We then calculated the centroid of these pixels, as the weighted average of their coordinates in both image directions, and the number of pixels over the significance threshold. Based on these two descriptors, we respectively describe the location and size of common regions of excitation across subjects for each trial.

Consistency across trials was assessed with the Binomial test. A binary image was created for each of the 12 trials, with 1's being attributed to pixels corresponding to the suprathreshold cluster in the pseudo t image for the observed data. These binary images were summed over classical excerpt trials and the octaves, separately for low and high playing speeds. Two resulting images were produced, each with pixel values ranging from 0 (under threshold across all trials) to 8 (over threshold across all trials). Considering a 50% chance that a pixel could be over threshold for each trial, the number (n) of pixels exceeding the significance threshold is 6, given that:

$$Pr(n > 6) = \sum_{i=0}^{8} Pr(n = i) = \sum_{i=0}^{8} \binom{8}{i} 0.5^8 < 5\% \text{ with } n \sim B(8, 0.5), \tag{1}$$

## Results

Notwithstanding the highly dynamic, playing conditions, all EMGs collected were of high quality (Fig 2C). Artifacts were occasionally observed in a few, non-consecutive electrodes. Nineteen out of 26,112 EMGs collected (96 electrodes x 34 trials x 8 subjects) were of low quality and were therefore interpolated. For one subject the most proximal probe run low on battery during acquisition and so no EMG was recorded for the Octaves, Sia and Rachmaninov trials. Permutations were limited to seven subjects for these trials.

For all trials, the number of 8-connected pixels exceeding the significance threshold ranged from 12 to 29 out of the 96 pixels in the grid. For the classical music excerpts, these suprathreshold pixels spanned the whole longitudinal forearm region covered by the grid, though only from roughly the 6th to the 8th decile of the forearm circumference (cf. white circles in Fig 4A). The region with consistently high EMG amplitude across subjects ranged from 12.5% to 20.8% of the whole size of the normalized grid, being centered roughly at 72% and 33% of the forearm circumference and length respectively (centroids in Fig 4A; see Fig 1B for anatomical references). For the octaves, the size of the region where suprathreshold pixels were found ranged from 25% to 30% of the normalized, grid size, being nearly 100% greater than that observed for the classical excerpts. Moreover, while the longitudinal centroid location was on average 32% of the forearm length, the centroid transverse location was confined from 9% to 13% of the forearm circumference (Fig 4B). Both the size of suprathreshold regions and the centroid coordinates changed by less than 2% when octaves were played with different speeds. When comparing the two groups of tasks, octaves and classical excerpts, EMGs were consistently represented at small regions relative to the grid size and centered at opposite forearm regions.

The consistency observed across subjects for the suprathreshold regions was not observed across trials. Suprathreshold pixels were either represented at the ventral or dorsal region, according to whether subjects respectively performed classical excerpts or octaves, regardless of the playing speed (Fig 5). None of the pixels provided a number of counts exceeding the significance threshold (n = 6) identified by the cumulative, Binomial distribution.

## Discussion

The objective of this study was to investigate the existence of forearm regions where EMGs with high amplitude could be consistently detected across expert piano players and conditions.

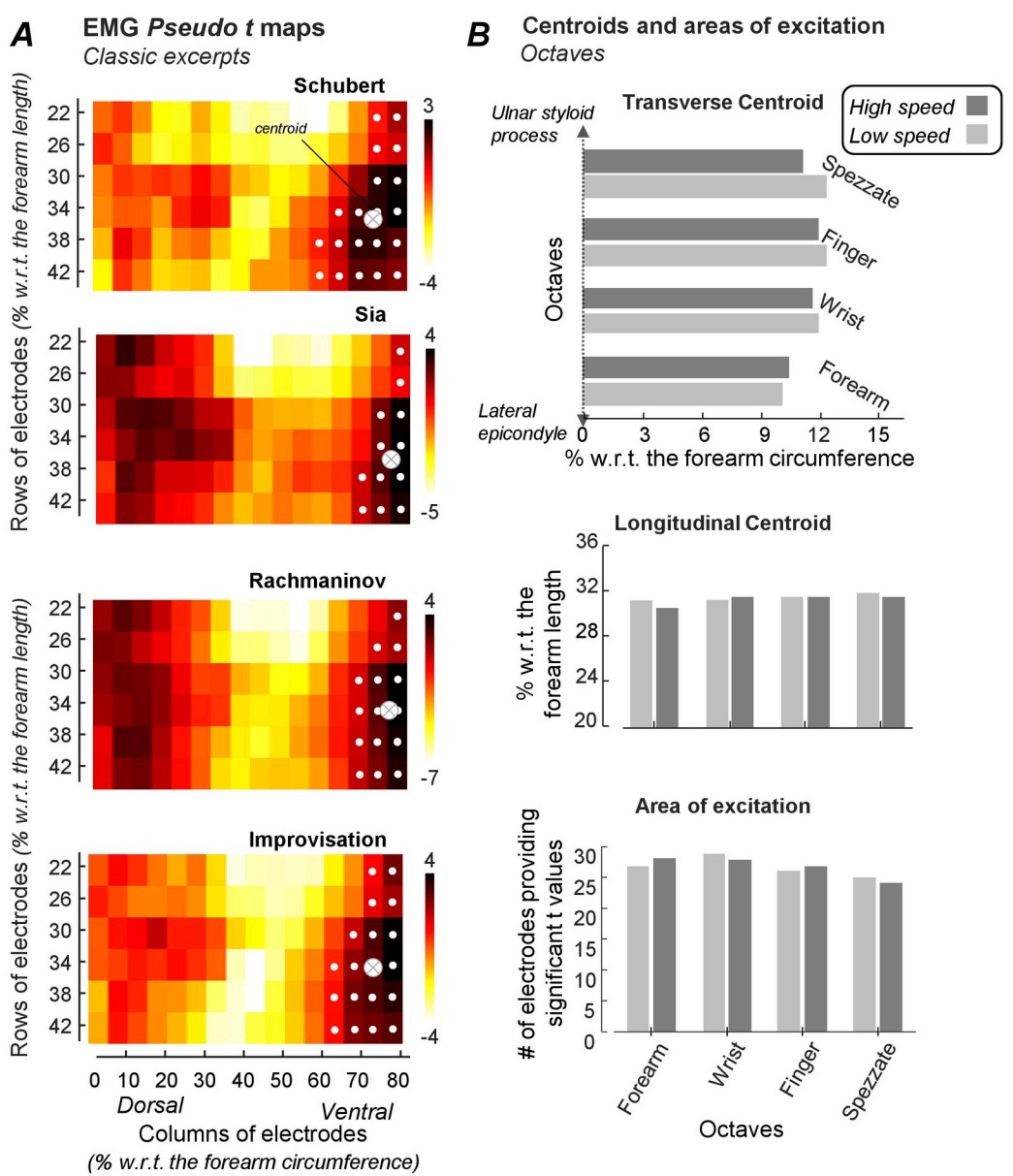

**Fig 4. Area and centroids of over-significance-threshold EMGs across subjects.** Pseudo t images obtained for classical excerpts are shown in A. White circles denote the region of excitation, defined by pixels providing an average pseudo t value greater than the statistical threshold, set at 5% (Fig 3, Step 4). Crossed circles indicate the centroid of the excitation region, computed as the weighted average of pseudo t values defining the region of excitation. Panel B shows transverse (up) and longitudinal (middle) coordinates of centroids during octaves, as well as the number of electrodes (bottom) defining the region of excitation.

Conditions were separated into two groups: octaves and classical excerpts. By identifying regions with consistently high EMG amplitude, interpreted as common regions of excitation of the forearm muscles, we expected to help furthering our understanding on the incidence of PRMDs and to provide an anatomical reference for the study of active muscle loading in piano players using surface EMG. Our key results revealed a consistency across subjects though not across tasks. Octaves and classical excerpts demanded the excitation of different forearm regions, which were however subject independent.

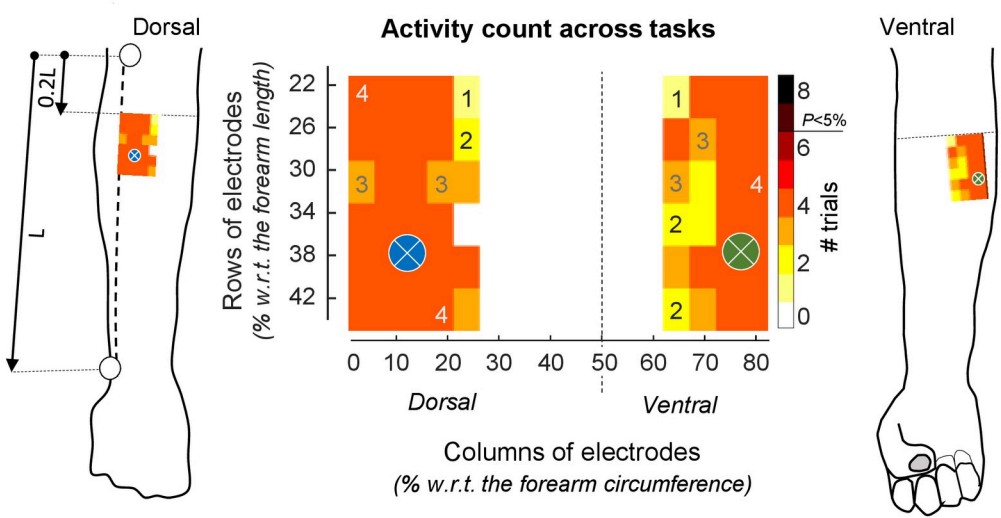

**Fig 5. Number of over-significance-threshold EMGs across trials.** Image showing the counts of pixels, across trials, providing pseudo t values which average was greater than the statistical threshold. Maximal count is eight, considering four classical excerpts and the four octaves performed at preferred speed. Counts are schematically represented with respect to anatomical references. Counts greater than 6 would indicate a significant consistency across trials. The exact same distribution of counts was obtained when considering octaves performed at high speeds. Regions where pairs of bipolar electrodes are recommended to be centered for assessing excitation of extensor (blue crossed circle) and flexor (green crossed circle) muscles are shown.

Two notes are warranted before discussing our results. First, we would like to emphasize our intention to assess where excitation is more likely to take place in the forearm rather than to compare changes in the degree of excitation between subjects and conditions. Our goal was motivated by the putative view that injuries are frequently associated with prolonged, active loading of skeletal muscles [25]. Considering pianists dedicate a large fraction of their time to continued practice [26], the high prevalence and incidence of PRMDs in the upper limbs are not surprising [7, 8]. Regardless of how much muscles may be excited, we therefore believe any forearm region being frequently excited across participants could posit a site where PRMDs are more likely to manifest. A second, technical note concerns the sensitivity of our detection system. Having a limited number of electrodes available (96 electrodes), we selected the detection system providing what we believe to be an appropriate compromise between forearm coverage and spatial resolution. Because of inter-individual variability, with our detection system (Fig 1) we were able to cover roughly 80% of the forearm circumference and 20% of the forearm length, with 16 and 6 detection points in both directions respectively. According to results reported during controlled, dynamic and isometric contractions [15, 16], the size and the density of the grid used here were expected to provide EMGs sensitive to excitation of the different, abutting muscles along and across the forearm.

Similarities in the region of excitation were observed when comparing octaves and classical excerpts. During selective contractions of fingers and wrist muscles, both in isometric and dynamic conditions, significant differences in the location where greatest EMGs were detected along the forearm have been reported [15, 16]. During piano playing, over pieces of classical excerpts and over a few seconds of octave performance, we however detected EMGs with significantly, consistently high amplitude along the whole forearm proximo-distal region covered by our grid of electrodes (Figs 1 and 4). This extensive proximo-distal region of excitation suggests different extensor or flexor muscles may be equally frequently elicited during piano

playing, which is in agreement with the variable kinematic demands imposed upon hand and finger movements during piano playing [27].

When considering the transverse, forearm regions, differences were noted between octaves and classical excerpts. For the octaves, the region consistently providing EMGs with high amplitude across subject was found dorsally in the forearm (Fig 4B). This result corroborates that reported by Furuya et al. [14] on expert pianists during octaves at different speeds. These authors obtained EMGs with higher amplitude from the extensor digitorum communis, located in the dorsal forearm region, when compared with those detected from the flexor digitorum superficialis, located ventrally in the forearm. Similarly, Oikawa et al. [19] also found higher EMG amplitude for the wrist extensors than flexors during octaves played at different speeds. Our results extend these previous observations, both to most of the dorsal forearm region and to octaves performed with different techniques: we detected EMGs with high amplitude over the whole proximo-distal region covered by electrodes placed dorsally on the forearm, for the four octaves and for the two playing speeds. More specifically, differences smaller than 2% on average were observed when comparing the size and location of the region of excitation between octaves (Fig 4B). This would seem to suggest that different octave techniques and tempo may demand a different degree of excitation, as reported in different studies [12–14], though not a differential, proximo-distal distribution of muscle excitation. This consistent, dorsal representation of EMGs was not observed however for classical excerpts. Different pieces of classical excerpts were selected to explore the greatest possible repertoire of movements required during piano playing. The three-minute improvisation (see S1 Appendix), in particular, was expected to introduce a large degree of variability between subjects, bearing in mind the different demands imposed on muscle excitation between random playing and sequential playing [20]. Strikingly, for the four classical excerpts we observed a consistent ventral representation of EMGs with high amplitude, extending proximo-distally along the grid (Fig 4A). Seemingly, the excitation of flexor muscles reported here is in disagreement with the notion of an efficient, playing style [28], whereby reliance on gravity is expected to allay the need of exciting wrist and finger flexor muscles. Being instructed to play as during a classical performance, it may be that subjects felt under pressure and therefore relied on an active key striking strategy, notwithstanding their level of expertise in piano playing. It is also possible that the weight of individual fingers is not sufficient to ensure a successful, passive key stroke, requiring therefore a minimal though frequent excitation of finger flexors. Moreover, the lack of a consistent dorsal representation of EMGs is as important as the consistent excitation of ventral, forearm muscles. While on one hand our results suggest expert piano players commonly load their flexor, forearm muscles during classical performance, on the other hand, our results posit that different subjects may lean on different strategies for key releasing. It seems therefore plausible to focus attention on different forearm transverse and longitudinal regions when studying muscle function during piano playing.

Postural differences imposed by octaves and melodic playing could further explain the differences we observed. On the one hand, octaves stand in need of shoulder and elbow joint movements while keeping the wrist relatively rigid. The need of a stable, rigid posture of the wrist might frequently require the excitation of the whole forearm extensors, which might explain our results for octaves tasks. On the other hand, melodic playing stand in need for independent finger movements and thus for a less rigid wrist posture than octaves. The independence of individual finger may however lead to a more variable reaction forces after pressing the keys, possibly imposing a demand for eccentric contraction of flexor muscles to make sure the wrist does not collapse into extension from the reaction force. Only through an integrated biomechanical and electromyographic analysis, at the level of individual keystrokes, it

would seem possible to assess the mechanisms underpinning the ventral, forearm excitation during classical, piano playing.

This study is expected to open new fronts for the assessment of active muscle loading in piano players. The regions of forearm excitation reported in Figs 4 and 5, for instance, may posit an anatomical reference for comparing different populations. Given spatio-temporal differences in the pattern of muscle excitation are typically reported when learning new motor tasks [29] and in the presence of pain [30, 31], the possibility of mapping EMG amplitude on the forearm would seem to provide a high discriminative power for distinguishing novice pianists or pianists with either chronic or acute musculoskeletal complications from healthy, expert players as those assessed here. It could be that, in expert players, forearm sites more likely to be injured or where pain mostly manifests are recruited less frequently. Conversely, novice players could rely more strongly on cocontraction of synergists and antagonist, forearm muscles, leading to broader regions of excitation than those reported here. The potential of grids of electrodes in revealing spatial inhomogeneities in muscle excitation has, indeed, been recently explored in piano players. In contrast with our results, Goubault et al [18] reported a significantly greater decrease of the median frequency of EMGs detected from extensor rather than flexor, forearm muscles in professional piano player when performing classical excerpts. Giving we observed more frequent excitation of flexor muscles across subjects when playing classical excerpts, reasonably, one could expect fatigue to be more likely to manifest in flexor than extensor muscles. A possible explanation for this disconnect between studies is the load sharing between and within muscles. In the study of Goubault et al [18], subjects played for a substantially longer time than our subjects did. During the relatively short playing time considered here, spatial inconsistencies in the excitation of extensor muscles across subjects could have been due to the redistribution of excitation among different extensor muscles, a mechanism sought to delay fatigue [32]. Alternatively, the excerpts considered here may have imposed less postural demands on the forearm, allowing subjects to frequently silence their extensor muscles. In spite of these differences, with grids of electrodes the active demand on the population of forearm muscles may be well tracked during piano playing. We believe, indeed, that spatial associations between forearm sites where playing-related injuries manifest and regions where forearm muscles are often excited in pianists suffering from PRMDs may be explored with high-density EMGs. Furthering this issue in future studies is expected to contribute to our understanding on the prediction and treatment of PRMDs in piano players.

One final implication we would like to mention regards the anatomical reference provided here for electrode positioning. By normalizing the position of electrodes with respect to anatomical landmarks, we were able to assess the distribution of muscle excitation over comparable regions across subjects. The existence of a common region of excitation, located at opposite extremes of the forearm circumference and spanning slightly over than 20% of the forearm length, appears to motivate the definition of a region where EMGs should be sampled when using grids of electrodes is not possible. More specifically, centering single pairs of electrodes at roughly 15% and 75% of the forearm circumference, 35% distally to the elbow joint (cf. crossed circles in Fig 5), would appear to maximize the probability of detecting muscle excitation during piano playing while accounting for differences in the size of different bipolar electrodes typically used for recording surface EMGs.

This study has some potential limitations. First, due to the fixed size of the electrode grid, the totality of the forearm circumference could not be covered for lager subjects: considerations drawn here apply to 80% of the forearm circumference. Because of this limited sampling, we were overtly unable to assess consistencies in excitation for the uncovered region. It should be noted though that if, on the one hand, EMGs were sampled from a limited forearm circumference, on the other hand, we were able to assess the distribution of electric potential

over similar, relative skin regions across subjects—monopolar EMGs were interpolated using spatial coordinates relative to subjects. Moreover, due to the limited number of grids, only the proximal part of the forearm has been sampled, covering roughly 20% of the forearm length. Even though this region seems to fully convey the distribution of excitation of forearm muscles during isometric contractions [15], further studies are necessary to assess proximo-distal differences in excitation associated with piano playing.

Finally, even though we have used a powerful statistical approach [23] to assess the consistency of excitation in EMG images, we acknowledge the excitation of forearm extensor muscles urges further consideration. Inter-individual differences seemingly explain the lack of a common, dorsal region of excitation during classical excerpts. Three out of the eight subjects tested, indeed, provided EMGs with high amplitude in similar, dorsal forearm locations, leading to remarkably though not significantly high pseudo t values (cf Fig 4A). While future studies may help understanding whether expert players rely on different excitation strategies for the dorsal, forearm muscles during classical playing, our results indicate classical and octaves playing impose a continued and spatially extensive demand respectively upon the flexor and extensor forearm muscles.

## Supporting information

**S1 Appendix. Experimental instructions.**
(DOCX)

## Author Contributions

**Conceptualization:** Alba Thio-Pera, Taian Vieira.

**Data curation:** Alba Thio-Pera.

**Formal analysis:** Alba Thio-Pera.

**Investigation:** Alba Thio-Pera, Taian Vieira.

**Methodology:** Alba Thio-Pera, Andrea Manzoni, Fabrizio D'Elia, Giacinto Luigi Cerone, Taian Vieira.

**Project administration:** Alba Thio-Pera, Cristina Bignardi, Taian Vieira.

**Software:** Alba Thio-Pera.

**Supervision:** Giacinto Luigi Cerone, Mara Terzini, Marco Gazzoni, Taian Vieira.

**Validation:** Taian Vieira.

**Visualization:** Taian Vieira.

**Writing – original draft:** Alba Thio-Pera.

**Writing – review & editing:** Matteo De Carlo, Andrea Manzoni, Fabrizio D'Elia, Giacinto Luigi Cerone, Giovanni Putame, Mara Terzini, Marco Gazzoni, Cristina Bignardi, Taian Vieira.

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
