## [Decision Letter · Decision Letter 0]

20 Dec 2021

PONE-D-21-36749Are the forearm muscles excited equally in different, professional piano players?PLOS ONE

Dear Dr. Thió i Pera,

Thank you for submitting your manuscript to PLOS ONE. After careful consideration, we feel that it has merit but does not fully meet PLOS ONE’s publication criteria as it currently stands. Therefore, we invite you to submit a revised version of the manuscript that addresses the points raised during the review process.

We look forward to receiving your revised manuscript.

Kind regards,

Jason Organ

Academic Editor

PLOS ONE

Journal Requirements:

Additional Editor Comments:

Both reviewers are enthusiastic about this manuscript. Please address all the reviewers' comments in your revision. Thank you for submitted a very interesting paper to PLOS One.

Reviewers' comments:

Reviewer's Responses to Questions

**Comments to the Author**

1. Is the manuscript technically sound, and do the data support the conclusions?

Reviewer #1: Yes

Reviewer #2: Yes

2. Has the statistical analysis been performed appropriately and rigorously? 

Reviewer #1: Yes

Reviewer #2: I Don't Know

3. Have the authors made all data underlying the findings in their manuscript fully available?

Reviewer #1: No

Reviewer #2: Yes

4. Is the manuscript presented in an intelligible fashion and written in standard English?

Reviewer #1: Yes

Reviewer #2: Yes

5. Review Comments to the Author

Reviewer #1: Thank you for the opportunity to review this manuscript, “Are the forearm muscles excited equally in different, professional piano players?”. This study tests the hypothesis that regions of strong forearm muscle activity during dynamic piano playing will be similar across a cohort of experienced pianists. Results are consistent with the hypothesis, showing pronounced dorsal (i.e., extensor) muscle activity when playing octaves and pronounced ventral (i.e., flexor) muscle activity when playing classical/jazz pieces and improvising.

I really enjoyed reviewing this, both as a biomechanist and an amateur pianist myself. I think the approach is interesting and that the methods are sound. Given the broad appeal of music outside of scientific circles alone, I expect this study could generate broad interest. However, I have a few concerns that should be addressed prior to publication.

MAJOR CONCERNS

The primary finding of this study is that professional pianists show strong muscle activity in the region of the forearm roughly corresponding to the medial and lateral epicondyles – in other words, roughly corresponding to the common flexor and extensor mass, respectively. My concern is that given the limited proximo-distal spread of the electrodes, I’d be surprised if the authors found anything else – i.e., for the proximal forearm region sampled, the most likely outcome would be high activity in the common flexor and extensor masses. I know there are technical difficulties in adding more electrodes, or – alternatively – tradeoffs between sampling breadth and resolution if the electrode cuff were spread across the entire forearm. Nonetheless, I think some discussion about the possible limitations of ONLY sampling in the proximal forearm would be warranted.

Additionally, the manuscript begins with a discussion of playing related musculoskeletal disorders (PRMDs), partly as a justification for undertaking the study. Is it possible, in the discussion, to relate the findings of this study with particular PRMDs? For instance, do pianists get tendinitis in the regions identified as major centers of muscle activity?

Data availability - The authors state that project data are within the manuscript and supporting information files. However, I did not see raw project data in either place. As stated in the PLOS Data Policy, “Authors are required to make all data underlying the findings described fully available, without restriction, and from the time of publication.”.

MINOR CONCERNS

(Lines 94-103) Was there any effort to gauge the musculoskeletal health of the participants (i.e., did they have a histology of PRMDs, or any current pain or other issues)?

(Line 117) I believe the correct English translation for spezzate would be “broken octaves”. Would this be an example of what the pianists were required to do? https://youtu.be/cUOQJ7ukYrk?t=545

(Lines 122-124) Were pianists given any instructions on tempo for the classical/jazz pieces, or just told to play as slow/fast as they wished? Also, the standard English translations for the classical pieces would be “Prelude in C-sharp minor of Rachmaninov” and “Impromptu Op. 90 No. 2 of Schubert” – Opus 9 is a collection of dances. Finally, isn’t Andrea Manzoni the composer of Schicksal in Arbeit?

(Lines 149-150) Please indicate here that the forearm was in pronated position when the length measurement was taken. I was trying to figure out why length was defined as lateral epicondyle to ulnar styloid process (i.e., diagonally across the forearm) until I looked at Figure 1 and saw the forearm was pronated.

(Discussion) In thinking about the different EMG results for octaves versus more melodic pieces, I wondered if the results in both cases said more about isometric/eccentric muscle contractions, rather than concentric muscle contractions per se. For instance, playing octave runs requires the pianist to use lots of shoulder and elbow joint movement while keeping the wrists relatively rigid. In this case, one might expect strong activity across the entirety of the forearm muscles to maintain a stable (isometric) position of the wrist. Conversely, playing melodic pieces requires lots of independent finger movement and a more supple wrist (though this varies of course, depending on the piece and the style). Nonetheless, I would expect strong eccentric flexor activity to make sure the wrist does not collapse into extension from the reaction force of pressing the keys. Perhaps a discussion along these lines could be added to the Discussion section of the manuscript.

(Supplemental Information) Please provide and English language translation of the instructions. I used Google translate, and it seemed to work well.

Reviewer #2: In this study, Thio-Pera and colleagues studied excitation in a grid of EMG electrodes of eight professional piano players playing a mix of carefully chosen pieces. I think this is a fascinating piece that may be of interest to diverse audiences (musicians, healthcare providers and anatomists), and therefore, PLoS ONE is a great journal choice. Although I believe that this piece is essentially publishable as is, I have one caveat and two suggestions:

Caveat: the statistics went WAY above my head. I really do not understand these analyses and have no clue what “pseudo t values” are. Although the results and discussion thereof seem plausible to me, if someone with more familiarity with these analytical methods calls them into question, I defer to that opinion.

Suggestion 1: I think that the figures in general could use some refinement. Specifically, Figure 1 needs more explanation and does not seem to fully agree with what is written about these methods in the text. I would request that the authors review the methodological description of electrode placement and make sure that it is clearly in accordance with this figure (e.g., I don’t really understand where any but the reference electrodes were placed or where the “circumference of the forearm was measured at 10% of its length”). If it is discussed in the methods, could the authors please show it in the figures? The other figures are even more confusing. Are these heat maps of one individual? Playing one piece? A combination of all of the individuals? Figure 3 is especially confusing. It seems that this is a whole explanation of a method but this is not spelled out really in the text or in enough detail in this figure. Although figure 5 is a bit clearer than most, it is a prime example of how the figure does not match the text: “centering single pairs of electrodes at roughly 15% and 75% of the forearm circumference, 35% distally to the elbow joint”. None of these values are actually shown as landmarks on the figure. This would be an easy addition that would very much help readers understand.

Suggestion 2: As an anatomist, I would really appreciate deeper discussion of the EMG results from an anatomical perspective – maybe even only one more discussion paragraph? Aside from broad statements about flexor vs. extensor compartments, could the authors discuss differences in medial or lateral activation? What is the significance of only looking at the proximal portion of the muscles? Might the results have been different at more distal placement of the grid? Speaking of the grid, could the authors add more to the caveat about the use of the static grid? These electrodes would have been measuring very different muscles in the player with the smallest forearm circumference vs. the one with the largest one. As the authors seemingly know, it would have been better to use a grid pattern that scaled the distance between electrodes based on forearm dimensions rather than a static pattern. This might have been achievable with some kind of elastic mount. Unfortunately, that is not the way the data were collected. A few more sentences about the effects of this error (more than just ‘we mostly missed the ulnar bone region in the big guys’) might help future investigators not repeat it.

With all of that said, I think it would be reasonable to publish this paper largely as it is. Yes, there are a couple of improvements that could be made (especially in retrospect, as seems to always be the case, right?), but this is an interesting addition to the literature that may be of value across disciplines.

Nice work! Adam Hartstone-Rose

6. PLOS authors have the option to publish the peer review history of their article (what does this mean?). If published, this will include your full peer review and any attached files.

Reviewer #1: **Yes: **Jesse W. Young

Reviewer #2: **Yes: **Adam Hartstone-Rose

---

## [Author Response · Author response to Decision Letter 0]

22 Jan 2022

Responses are attached at the end of this PDF.

---

## [Decision Letter · Decision Letter 1]

4 Mar 2022

Are the forearm muscles excited equally in different, professional piano players?

PONE-D-21-36749R1

Dear Dr. Thió i Pera,

We’re pleased to inform you that your manuscript has been judged scientifically suitable for publication and will be formally accepted for publication once it meets all outstanding technical requirements.

Kind regards,

Jason Organ

Academic Editor

PLOS ONE

Additional Editor Comments (optional):

Reviewers' comments:

Reviewer's Responses to Questions

**Comments to the Author**

1. If the authors have adequately addressed your comments raised in a previous round of review and you feel that this manuscript is now acceptable for publication, you may indicate that here to bypass the “Comments to the Author” section, enter your conflict of interest statement in the “Confidential to Editor” section, and submit your "Accept" recommendation.

Reviewer #1: All comments have been addressed

Reviewer #2: All comments have been addressed

2. Is the manuscript technically sound, and do the data support the conclusions?

Reviewer #1: Yes

Reviewer #2: Yes

3. Has the statistical analysis been performed appropriately and rigorously? 

Reviewer #1: Yes

Reviewer #2: I Don't Know

4. Have the authors made all data underlying the findings in their manuscript fully available?

Reviewer #1: Yes

Reviewer #2: Yes

5. Is the manuscript presented in an intelligible fashion and written in standard English?

Reviewer #1: Yes

Reviewer #2: Yes

6. Review Comments to the Author

Reviewer #1: The authors have done a great job addressing my comments and those of the other reviewers. I recommend publication.

Reviewer #2: Having found the original version of this paper nearly publishable as it was, and being satisfied that the authors adequately considered the feedback that both of us reviewers gave, I have no hesitation in recommending publication of the current state of the manuscript (unless the other reviewer and/or editor still required minor modifications). This is an interesting paper. As a pure anatomist, it is interesting to think about the muscles that I study not only from an electrophysiological perspective (something that many of my colleagues have done, though I have not), but also from the perspective of how these muscles function in essentially elite use. Clearly we did not evolve for virtuoso piano playing, but were essentially exapted for this behavior and understanding the depth of that ability from a physiological perspective is indeed intriguing.

Again, this is nice work! Sincerely, Adam Hartstone-Rose

7. PLOS authors have the option to publish the peer review history of their article (what does this mean?). If published, this will include your full peer review and any attached files.

Reviewer #1: **Yes: **Jesse W. Young

Reviewer #2: No

---

## [Editor Report · Acceptance letter]

12 Mar 2022

PONE-D-21-36749R1 

Are the forearm muscles excited equally in different, professional piano players? 

Dear Dr. Thio-Pera:

I'm pleased to inform you that your manuscript has been deemed suitable for publication in PLOS ONE. Congratulations! Your manuscript is now with our production department. 

Kind regards, 

on behalf of

Dr. Jason Organ 

Academic Editor

PLOS ONE